# Attitude and Performance for Online Learning during COVID-19 Pandemic: A Meta-Analytic Evidence

**DOI:** 10.3390/ijerph191912967

**Published:** 2022-10-10

**Authors:** Xuerong Liu, Zheng Gong, Kuan Miao, Peiyi Yang, Hongli Liu, Zhengzhi Feng, Zhiyi Chen

**Affiliations:** 1School of Psychology, Third Military Medical University, Chongqing 400038, China; 2Experimental Research Center for Medical and Psychological Science (ERC-MPS), Third Military Medical University, Chongqing 400038, China

**Keywords:** COVID-19, online learning, learning attitude, learning performance

## Abstract

The COVID-19 pandemic prominently hit almost all the aspects of our life, especially in routine education. For public health security, online learning has to be enforced to replace classroom learning. Thus, it is a priority to clarify how these changes impacted students. We built a random-effect model of a meta-analysis to pool individual effect sizes for published articles concerning the attitudes and performance towards online learning. Databases included Google Scholar, PubMed and (Chinese) CNKI repository. Further, a moderated analysis and meta-regression were further used to clarify potential heterogenous factors impacting this pooled effect. Forty published papers (n = 98,558) were screened that were eligible for formal analysis. Meta-analytic results demonstrated that 13.3% (95% CI: 10.0–17.5) of students possessed negative attitudes towards online learning during the COVID-19 pandemic. A total of 12.7% (95% CI: 9.6–16.8) students were found to report poor performance in online learning. Moderated analysis revealed poor performance in online learning in the early pandemic (*p* = 0.006). Results for the meta-regression analysis showed that negative attitudes could predict poor learning performance significantly (*p* = 0.026). In conclusion, online learning that is caused by COVID-19 pandemic may have brought about negative learning attitudes and poorer learning performance compared to classroom learning, especially in the early pandemic.

## 1. Introduction

Coronavirus Disease 2019 (COVID-19), a pandemic that is caused by Severe Acute Respiratory Syndrome Coronavirus 2 (SARS-CoV-2) transmission, has brought about unprecedented losses in the global sphere. The World Health Organization (WHO) officially recommended the lockdown policy to disrupt virus transmission and slow the development of COVID-19 pandemic [1]. It prominently hit almost all the aspects of our life, especially in routine education. The United Nations Educational, Scientific and Cultural Organization (UNESCO) (2022) provided influential reports that the COVID-19 pandemic disrupted the schooling of more than 1.6 billion students. To make matter worse, school closes led to 91.3% of total enrolled learners in 188 countries at all levels of learning losing social-interactive education [2]. To control social distance, policymakers strongly recommended to cease face-to-face classroom lessons for all the educational institutes [3]. On the other hand, as one of the most common compensatory measures, cyber (online) teaching has been utilized to substitute for the traditional teaching model. Though a timely and potential replacement, we are still unclear as to whether such changes bring about an unexpected impact on educational outcome.

Online learning (i.e., cyber learning, e-learning, digital learning, or computer-based learning) refers to a form of remote-teaching by using a digital device and electric teaching materials [4]. Compared to face-to-face classroom learning, online learning was found to have higher flexibility and availability [5,6,7]. Thus, online learning has proliferated in the COVID-19 pandemic, which is found to have several prominent strengths, such as high learning satisfaction and substantial educational resources. On the other hand, concerns for online learning are continuing, especially concerning poor real-time feedback, less social interaction and uncontrolled situations [8,9]. Supporting this, existing studies have demonstrated the close association of online learning with poor learning performance during the lockdown period [10,11,12]. Thus far, debates for the advantages (and also disadvantages) of online learning are ongoing [7,13,14]. Almost all education has been converted into an online learning form in the world currently. Thus, what the influences of such large-scale online learning are on educational outcomes should be clarified.

A major concern for online learning is the negative attitude towards such learning techniques [15,16,17]. Though having potential, huge fruits from online learning are reaped for not all the students given the technical gap [18,19,20,21]. In other word, enforcing students into online learning, caused by COVID-19, may aggravate rich–poor polarities, so as to disrupt educational equities [22,23,24,25]. There were considerable findings for the close association between learning attitude and learning performance [26,27,28]. Iqra and colleague (2021) demonstrated that negative attitudes for online learning technology, study environment and teaching interaction prospectively predicted poor performance [29]. Thus, probing the influence of online learning attitudes on performance could be beneficial for policymakers to rethink educational measures during the COVID-19 pandemic.

To clarify students’ attitudes towards online learning and corresponding learning performance, we performed a meta-analysis to investigate the rate of negative attitudes for online learning. Further, the same pipeline was used to examine the rate of learning performance for online learning. Targeted literature have been retrieved in databases including Google Scholar, PubMed, EMBASE, Cochrane Library, EBSCO, Web of Science and CNKI (Chinese database) on 20 June 2022. Moreover, to gain further information, we capitalized on a moderation analysis for the hierarchical effect of educational phases and lockdown period on the meta-analytical effect. Finally, we have conducted meta-regression to examine the potential association between negative attitudes towards online learning and learning performance.

## 2. Materials and Methods

To improve reproducibility, all the pipelines and protocols were followed by the Cochran Handbook and Preferred Reporting Items for Systematic Reviews and Meta-Analyses (PRISMA) [30] statement (see Figure 1). All the raw data, scripts and materials have been submitted into the Open Science Framework (OSF) repository.

### 2.1. Search Strategy and Selection Criteria

Literature searching was conducted for studies concerning attitudes and performance towards online learning during the COVID-19 pandemic (February 2020–June 2022). Retrieval databases included Google Scholar, PubMed, EMBASE, the Cochrane Library, EBSCO, Web of Science and CNKI. We capitalized on Boolean logic for literature searching in databases as aforementioned: (e-learning OR online learning) AND (COVID-19) AND (attitude OR satisfaction) AND (learning effect OR learning performance) NOT (review OR meta analysis). Inclusion criteria are as follows: (1) quantifiable measures for attitudes or performance towards online learning; (2) sampling duration should be limited to the COVID-19 pandemic; (3) fundamental metainformation should be given, such as sample size and sample population. Studies were excluded by the following criteria: (1) review, perspective, opinion or meta-analytic results; (2) no quantifiable investigation or data. For moderation analysis, the sample population has been partitioned into two groups based on educational level: primary–secondary school students and college students. Further, we coded the pandemic period into two phases, with February 2020–December 2020 being the early stage in COVID-19 pandemic and with January 2021–June, 2022 being the later stage in the pandemic.

### 2.2. Data Extraction and Quality Assessment

Data were extracted from eligible articles by two extractors independently, including sample size, sampling population, demographic information, educational level, pandemic period, sampling time, the percentage of expressing negative evaluation for online learning, and the percentage of reporting poor learning performance in online learning. All the data have undergone cross-validation between two independent extractors. For quality control, two additional assessors independently evaluated the evidence (study) quality by using the modified Newcastle-Ottawa Scale (mNOS) [31,32]. Potential disagreements were resolved by a third assessor author. Specifically, the quality assessment evaluation criteria of mNOS were as follows: sample misrepresentations and size; comparability between respondents and non-respondents; ascertainment of online learning attitude and study performance; sufficient adequacy of the descriptive statistics. The total quality scores ranged from 0 to 5; the low risk of bias as expressed by the mNOS was ≥3 points, whilst the high risk of bias was ranked based on low mNOS scores (<3 points) (Appendix A).

### 2.3. Encoding and Statistical Analysis

Meta-analyses were implemented by using Comprehensive Meta-Analysis (CMA) software (V3, Biostat Inc., Englewood, CO, USA). To estimate within-study and between-study heterogeneity in meta-analytic models, the I^2^ test and Cochran’s Q test were used. As indicated in previous studies, meta-analysis with a fixed-effect model was suitable for estimating the meta-analytic effect with no heterogeneity (I^2^ < 50%, *p*-value ≥ 0.1) [33], while a random-effects model should be used to adjust high data within-study and between-study heterogeneity (I^2^ > 50%, *p*-value < 0.1). In the current study, the random-effect meta-analytic models were built to examine the attitudes and performance towards online learning during COVID-19 pandemic due to high heterogeneity (see Results section). In addition, to validate whether these meta-analytic effects are moderated by potential factors (i.e., educational level and pandemic period), the moderated-effect models were constructed by labeling individual effect sizes into subgroups (i.e., primary–secondary school group and college group; early pandemic group (February 2020–December 2020) and late group (January 2021–June 2022)). Furthermore, for testing the potential role of attitudes on performance for online learning, we built a meta-regression model by fitting the individual effect size of attitudes to the performance for all the included studies. Finally, as quality control, funnel charts were generated for visually inspecting publication bias in the current study, with asymmetric distribution for high publication bias. In addition, the Kendall’s test and Egger’s test were also used for quantitative validation for publication bias, with *p* < 0.05 indicating significant publication bias.

## 3. Results

In the current study, 40 papers were screened from 856 papers for the final data pool in the meta-analysis (*n* = 98, 558). Sample populations covered China, Indonesia, American, Nepal, Sri Lanka and Canada (see Table 1).

### 3.1. Heterogeneity Test

Results for the heterogeneity test showed the high between-study heterogeneity (I^2^ > 98%, *p* < 0.05), which indicated that the random effect model should be used for the pooling effect size of included studies. In addition, high heterogeneity implied potential moderators.

### 3.2. Main Meta-Analysis

We coded attitudes for online learning from the included studies into three categories that included positive attitude, uncertain attitude and negative attitude. Further, subjective learning performance was categorized into poor, uncertain and high. Meta-analytic results showed a statistically significant negative attitude for online learning during COVID-19 (13.3%, *p* = 0.000, 95% CI: 10.0–17.5; N = 95,654). In addition, there was poor study performance (12.7%, *p* = 0.000, 95% CI: 9.6–16.8; N = 20,528). Finally, only 21% (*p* = 0.000, 95% CI: 12.3–34.0; N = 40,573) of students reported that online learning was better than face-to-face classroom learning (see Figure 2).

### 3.3. Moderation Analysis

Here, we used moderation analysis for examining the moderated role of educational levels (primary–secondary schools students and college students) and pandemic period (early stage of lockdown and later stage of lockdown) on the meta-analytic effect. Results demonstrated a null finding for the moderated effect of educational level on the negative attitude towards online learning and learning performance. In addition, there were no significant moderated effects of the pandemic period on negative attitudes towards online learning. Interestingly, we found that pandemic period significantly moderated learning performance for online learning, with poor performance for the early stage (Q-value = 7.615, *p*-value = 0.006).

### 3.4. Meta-Regression Analysis

We further conducted a meta-regression analysis to examine the association between the negative attitudes towards online learning and learning performance. Results demonstrated that the negative attitudes could predict poor learning performance significantly (β = 0.49, (95% CI: −0.006–1.127), t = 2.12, *p* = 0.026 (one-tailed)).

### 3.5. Publication Bias Assessment

Funnel plots were used for inspecting publication bias visually. The symmetrical distribution was found for z-scores of standard error among these studies. Thus, no significant publication was perceived. Further, Begg’s rank test was performed to quantitatively test the publication bias. Results showed no publication biases for both meta-analyses (negative attitudes for online learning, Kendall’s tau = −0.032, *p* = 0.784, continuity-corrected; learning performance for online learning, Kendall’s tau = −0.094, *p* = 0.576) (Appendix A).

## 4. Discussion

To investigate what the proliferation of online learning has brought about for educational outcomes during the COVID-19 pandemic (January 2020–June 2022), we performed a meta-analysis concerning attitudes and learning performance for online learning. Result showed that 13.3% of students possessed statistically significant negative attitudes for online learning during the COVID-19 pandemic. In addition, 12.7% students were found to report a poorer learning performance in online learning when compared to face-to-face classroom learning. Moreover, we found that the pandemic period significantly moderated the meta-analytic effect of learning performance on online learning, with poorer learning performance in the early pandemic (i.e., Januray 2020–January 2021). Finally, results for meta-regression analysis showed that the negative attitudes towards online learning could predict poor learning performance significantly. In conclusion, these finding indicated that neither learning attitudes nor learning performances were poorer for large-scale online learning compared to face-to-face classroom learning. Thus, despite the replacement, the consistent online learning that is enforced by public health policy may incur poor educational quality.

Main findings in the current study revealed that 13.3% students expressed a negative attitude for online learning during the COVID-19 pandemic. Furthermore, 12.7% students reported poor learning performance in online learning compared to face-to-face classroom learning. Although it seems to be a small portion of students, the vast majority of students still reported no significant improvement or satisfaction for online learning (e.g., “I am not certain whether I feel satisfactory for online learning”). This situation may be explained partly by two reasons: social isolation and learning resource gaps (see Table 2).

Social isolation has long been acknowledged as one of the best barriers to impede teaching performance in online learners [74,75,76]. Several lines of evidence demonstrate that the social isolation between teachers and learners could undermine learning satisfaction, and even make learners prone to experience negative emotions, such as anxiety, depression and loneliness [77,78]. Though lockdown policy obstructed virus transmission, it is of obvious consequence that the real-world social interaction between students and teachers has been curbed. Thus, increasing social interaction in online learning may be promising to control the negative impact of online learning on educational outcomes.

On the other hand, existing evidence was provided to indicate the potential impact of learning resource gaps on learning satisfaction and fairness during the COVID-19 pandemic [79,80]. Specifically, when compared to middle- and low-income countries (areas), students in high- or upper-middle-income countries (areas) rated online learning as more positive due to superior learning resources [81,82]. Supporting that, further studies revealed that students with disadvantaged positions economically would receive less high-quality learning resources, so as to incur poor academic performance [83,84,85]. Thus, this indirect evidence may provide another insight to warrant stakeholders for the potential risk of this change (i.e., enforcing online learning) in exacerbating educational inequality. In addition, it is remarkable that this conjecture should be further validated elsewhere and lacking direct evidence in the current study.

Another key finding worthy to discuss was that the pandemic period significantly moderated the meta-analytic effect of learning performance for online learning, with poorer performance in the early stage. There is a lot of evidence that the inadequate preparation and experience for online learning was partly responsible for this outcome. In the early stage of pandemic, students have been found to face a lot problems in participating in online courses, technical gaps, out-of-control class quality and social isolation [70,71,72,86]. Fortunately, with the development of user-friendly tools and increased experiences, technical issues and social interaction between teachers and students in online have been increasingly addressed [15,87]. Therefore, despite the presence of negative impacts, it had been controlling for online learning during COVID-19 pandemic.

Ultimately, by using meta-regression analysis, we found that the negative attitude towards online learning could predict poor learning performance. It had long been acknowledged that attitudes are composed of emotion, behavior and cognitive components, which could direct ones’ behavior. A robust body of studies have demonstrated the close association between learning attitudes and learning performance [27,88,89,90,91,92]. Furthermore. neuroimaging-based evidence supported this point by showing the hyper-activation of the hippocampus (i.e., the core brain region for working memory) in resolving n-back-like mathematical tasks with evoked positive emotions (L. Chen et al., 2018).

In spite of its strengths, this study still has several limitations. One shortcoming worthy to note is that these conclusions should be extended warily as we used biased sample populations. Thus, it may be worthy to extend sample representativeness in the future. Secondly, learning performance claimed in the current study referred to self-reported subjective evaluations. In other words, extending this conclusion elsewhere should cautiously used due to the gap of self-reported evaluation and actual performance. Lastly, to provide robustly statistical evidence, we had to remove a certain number of relevant studies because of lacking standardized or quantitative measures. Thus, the total sample size may be limited in the current study. The majority of the included studies examined one’s attitudes and learning performance by using signaling questions instead of standardized questionnaire or scales. Thus, future studies could provide evidence more robust by using valid and reliable measures.

## 5. Conclusions

The COVID-19 pandemic attacked almost all aspects of our life, especially in routine educational activities. To tackle this crisis well, the mainstream educational pattern has changed from face-to-face classroom learning into virtual online learning (e.g., webinar or iCloud classroom). However, this still lacks a systematic review and meta-analysis to make clear how this change impact our educational outcome. To clarify this issue, we conducted a systematic meta-analysis for literature concerning the attitudes and learning performance towards online learning during the COVID-19 pandemic. Results showed that 13.3% of students possessed statistically significant negative attitudes towards online learning during COVID-19 pandemic. In addition, 12.7% students were found to report a poorer learning performance in online learning compared to face-to-face classroom learning. Further, this meta-analytic effect was significantly moderated by the pandemic period, with poorer learning performance in the early stage than the latter one. Lastly, by using meta-regression, we revealed that the learning performance could be predicted significantly by learning attitudes towards online learning. On the balance, these findings allow us to draw the conclusion that the online learning that has been caused by COVID-19 pandemic may have brought about negative attitudes and promoted decreased learning performance compared to face-to-face classroom learning, especially in the early stage.

## Figures and Tables

**Figure 1 ijerph-19-12967-f001:**
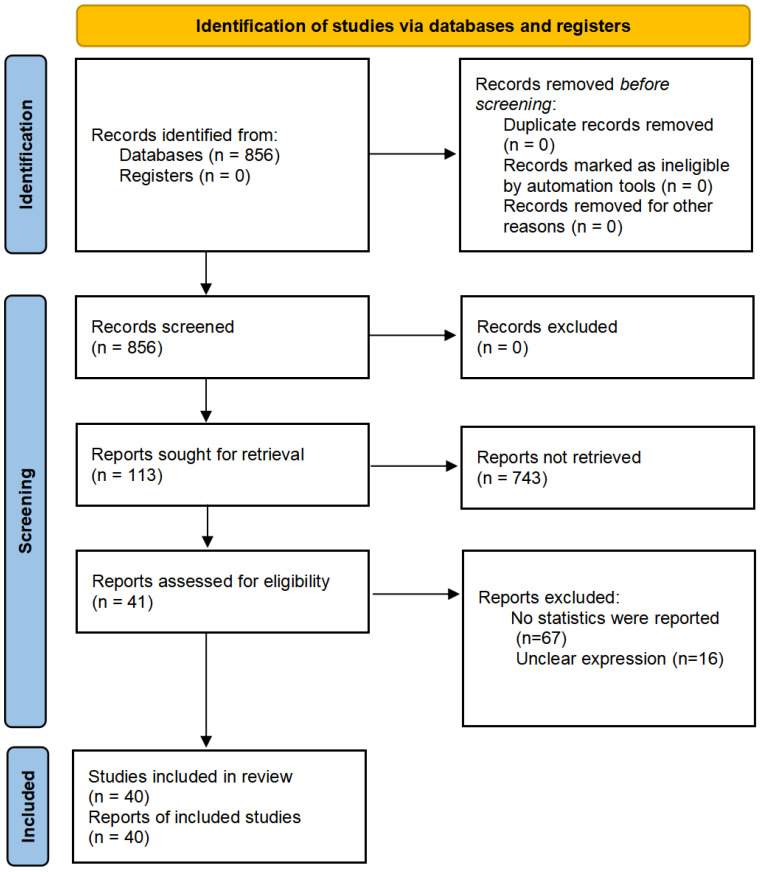
Flow chart of the study selection process in the 2020 PRISMA protocol. This flowchart coincides with the broad-certified 2020 PRISMA statement.

**Figure 2 ijerph-19-12967-f002:**
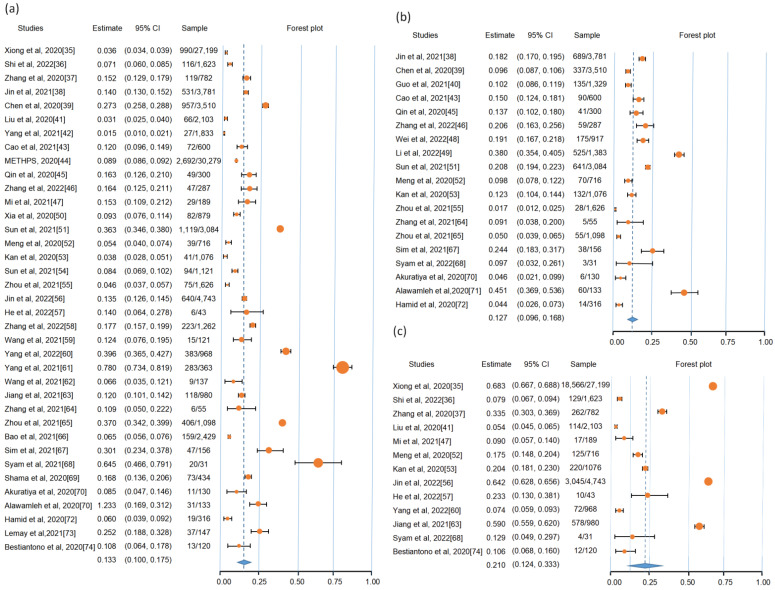
Forest plot for meta-analytic results. (**a**) The negative attitude for online learning during COVID-19 pandemic [34,35,36,37,38,39,40,41,42,43,44,45,46,47,48,49,50,51,52,53,54,55,56,57,58,59,60,61,62,63,64,65,66,67,68,69,70,71,72,73]; (**b**) The poor study performance of online learning [37,38,39,42,44,45,47,48,50,51,52,54,63,64,66,67,68,69,70,71]; (**c**) Online learning was better than face-to-face classroom learning [34,35,36,40,46,51,52,55,56,59,62,67,73]. The circle colored by orange represents the point estimation for effect towards corresponding study, with the large circle size representing a high effect size.

**Table 1 ijerph-19-12967-t001:** Summary of characteristics of included studies. N.A. = not applicable.

Author	Publication Time	Data Collection Time	Region	Sample Size	Gender		Education Phase	Negative Attitude of Online Learning		Poor Learning Performance		Online Learning Better than Face-To-Face Classroom Learning	
					Male	Female		Number	Rate	Number	Rate	Number	Rate
(Xiong and Wu, 2020) [34]	2020.08.17	NA	CN	27,199	13,691	13,508	primary and secondary school	990	3.64%	NA	NA	18,566	68.26%
(Shi et al., 2022) [35]	2022.1	2020.3	CN	1623	154	1469	college	116	7.15%	NA	NA	129	7.95%
(L. Zhang et al., 2020) [36]	2020.3	NA	CN	782	NA	NA	college	119	15.22%	NA	NA	262	33.50%
(Y. Jin et al., 2021) [37]	2021.4	2020	CN	3781	1950	1831	college	531	14.04%	689	18.22%	NA	NA
(J. Chen and Cui, 2020) [38]	2020.11	2020.5	CN	3510	1267	2243	college	957	27.26%	337	9.60%	NA	NA
(Guo and Wen, 2021) [39]	2022	2020.3	CN	1329	76	1253	college	NA	NA	135	10.16%	NA	NA
(Y. Liu and Zhang, 2020) [40]	2020.9	NA	CN	2103	702	1404	college	66	3.12%	NA	NA	114	5.40%
(H. Yang and Li, 2021) [41]	2022.4	NA	CN	1833	668	1165	college	27	1.46%	NA	NA	NA	NA
(Cao, 2021) [42]	2021.4	2020.6	CN	600	228	372	primary and secondary school	72	12.00%	90	15.00%	NA	NA
(Ministry Of Education and The Chinese Physical Society, 2020) [43]	2020.3.25	NA	CN	30,279	NA	NA	college	2692	8.89%	NA	NA	NA	NA
(Qin et al., 2020) [44]	2020	NA	CN	300	41	259	college	49	16.48%	41	13.78%	NA	NA
(H. Zhang et al., 2022) [45]	2022	2020.10–2021.4	CN	287	56	231	college	47	16.38%	59	20.56%	NA	NA
(Mi et al., 2021) [46]	2021.3	NA	CN	189	90	99	college	29	15.31%	NA	NA	17	8.99%
(Wei et al., 2022) [47]	2022.1.26	NA	CN	917	179	738	college	NA	NA	175	19.12%	NA	NA
(Juntang Li et al., 2022) [48]	2021.6	2020.9	CN	1383	NA	NA	college	NA	NA	525	37.96%	NA	NA
(Xia, 2020) [49]	2020.1	NA	CN	879	489	390	primary and secondary school	82	9.30%	NA	NA	NA	NA
(S. Sun and Wu, 2021) [50]	2021.8	NA	CN	3084	1618	1466	college	1119	36.30%	641	20.80%	NA	NA
(Meng et al., 2020) [51]	2020.3.27	NA	CN	716	NA	NA	college	39	5.50%	70	9.80%	125	17.50%
(Kan et al., 2020) [52]	2020.6	NA	CN	1076	451	625	college	41	3.81%	132	12.27%	220	20.45%
(X. Sun et al., 2021) [53]	2021.2	NA	CN	1121	535	586	college	94	8.38%	NA	NA	NA	NA
(Zhou, 2021) [54]	2021	NA	CN	1626	804	822	college	75	4.61%	28	1.72%	NA	NA
(Z. Jin et al., 2022) [55]	2022.5	2022.4	CN	4743	1281	3462	college	640	13.50%	NA	NA	3045	64.20%
(He et al., 2022) [56]	2022.4	NA	CN	43	NA	NA	college	6	13.64%	NA	NA	10	22.70%
(Y. Zhang et al., 2022) [57]	2020.11	NA	CN	1262	NA	NA	college	223	17.68%	NA	NA	NA	NA
(B. Wang et al., 2021) [58]	2021	NA	CN	121	NA	NA	college	15	12.40%	NA	NA	NA	NA
(Y. Yang, 2022) [59]	2022.6	2020.6	CN	968	101	867	college	383	39.56%	NA	NA	72	7.44%
(M. Yang and Qiu, 2021) [60]	2021.4	NA	CN	363	187	176	primary and secondary school	283	78.02%	NA	NA	NA	NA
(Y. Wang, 2021) [61]	2021.6	NA	CN	137	68	69	primary and secondary school	9	6.57%	NA	NA	NA	NA
(Jiang, 2021) [62]	2021.6	2020.5	CN	980	481	499	primary and secondary school	118	12.00%	NA	NA	578	59.00%
(X. Zhang, 2021) [63]	2021.5	NA	CN	55	27	28	primary and secondary school	6	11.00%	5	9.00%	NA	NA
(Zou, 2021) [64]	2021.4	2020.7	CN	1098	NA	NA	college	406	37.00%	55	5.04%	NA	NA
(Bao, 2021) [65]	2021.5	2020.4	CN	2429	1211	1218	primary and secondary school	159	6.55%	NA	NA	NA	NA
(Sim et al., 2021) [66]	2021.1.25	NA	Hongkong	156	57	99	college	47	30.13%	38	24.36%	NA	NA
(Syam and Achmad, 2022) [67]	2022.2.28	NA	Indonesia	31	NA	NA	college	20	65.60%	3	9.40%	4	12.50%
(Sharma et al., 2020) [68]	2020	NA	Nepal	434	97	337	college	73	16.82%	NA	NA	NA	NA
(Akuratiya and Meddage, 2020) [69]	2020	NA	Sri Lanka	130	67	63	college	11	8.50%	6	4.60%	NA	NA
(Alawamleh et al., 2020) [70]	2020.8.10	NA	American	133	70	63	college	31	23.30%	60	45.11%	NA	NA
(Hamid et al., 2020a) [71]	2020.6.18	2020.5	Indonesia	316	NA	NA	college	19	5.90%	14	4.40%	NA	NA
(Lemay et al., 2021) [72]	2021.6.29	NA	Canada	147	66	81	college	37	25.19%	NA	NA	NA	NA
(Bestiantono et al., 2020) [73]	2020.12	NA	Indonesia	120	60	60	primary and secondary school	13	11.10%	NA	NA	12	10.30%

**Table 2 ijerph-19-12967-t002:** Summary for self-reported reasons to negatively evaluate online learning from previous studies.

Reason	Frequency	Reference
Learning resources (lack of learning materials, technical support, learning platform, learning equipment. limited access to the internet through a mobile phone or other deice)	28	(Xiong and Wu, 2020 [34]), (Shi et al., 2022 [35]), (J. Chen and Cui, 2020 [38]), (Guo and Wen, 2021 [39]), (Y. Liu and Zhang, 2020 [40]), (H. Yang and Li, 2021 [41]), (Ministry Of Education and The Chinese Physical Society, 2020 [43]), (Qin et al., 2020 [44]), (Mi et al., 2021 [46]), (Juntang Li et al., 2022 [48]), (Xia, 2020 [49]), (X. Sun et al., 2021 [53]), (Zhou, 2021 [54]), (Y. Zhang et al., 2022 [57]), (B. Wang et al., 2021 [58]), (Y. Yang, 2022 [59]), (M. Yang and Qiu, 2021 [60]), (Y. Wang, 2021 [61]), (Jiang, 2021 [62]), (Zou, 2021 [64]), (Bao, 2021 [65]), (Sim et al., 2021 [66]), (Syam and Achmad, 2022 [67]), (Sharma et al., 2020 [68]), (Akuratiya and Meddage, 2020 [69]), (Hamid et al., 2020a [71]), (Lemay et al., 2021 [72]), (Bestiantono et al., 2020 [73])
Social isolation	21	(Xiong and Wu, 2020 [34]), (Shi et al., 2022 [35]), (J. Chen and Cui, 2020 [38]), (Y. Liu and Zhang, 2020 [40]), (H. Yang and Li, 2021 [41]), (Ministry Of Education and The Chinese Physical Society, 2020 [43]), (Qin et al., 2020 [44]), (Mi et al., 2021 [46]), (Xia, 2020 [49]), (Zhou, 2021 [54]), (He et al., 2022 [56]), (Y. Zhang et al., 2022 [57]), (Y. Yang, 2022 [59]), (M. Yang and Qiu, 2021 [60]), (Jiang, 2021 [62]), (Sim et al., 2021 [66]), (Syam and Achmad, 2022 [67]), (Akuratiya and Meddage, 2020 [69]), (Alawamleh et al., 2020 [70]), (Lemay et al., 2021 [72]), (Bestiantono et al., 2020 [73])
Teacher behavior (teaching ability, preparation, supervision and teaching guide)	11	(Xiong and Wu, 2020 [34]), (Shi et al., 2022 [35]), (Y. Liu and Zhang, 2020 [40]), (H. Yang and Li, 2021 [41]), (B. Wang et al., 2021 [58]), (Y. Yang, 2022 [59]), (M. Yang and Qiu, 2021 [60]), (Y. Wang, 2021 [61]), (Jiang, 2021 [62]), (Zou, 2021 [64]), (Hamid et al., 2020a [71])
Learning autonomy	9	(Xiong and Wu, 2020 [34]), (Xia, 2020 [49]), (S. Sun and Wu, 2021 [50]), (Zhou, 2021 [54]), (He et al., 2022 [56]), (Y. Yang, 2022 [59]), (M. Yang and Qiu, 2021 [60]), (Syam and Achmad, 2022 [67]), (Lemay et al., 2021 [72])
Teaching contents (course arrangement, course design, course time and classroom activities)	8	(Y. Liu and Zhang, 2020 [40]), (H. Yang and Li, 2021 [41]), (Zhou, 2021 [54]), (Y. Zhang et al., 2022 [57]), (Y. Yang, 2022 [59]), (M. Yang and Qiu, 2021 [60]), (Y. Wang, 2021 [61]), (Jiang, 2021 [62])

## Data Availability

All the materials regarding this study have been deposited in Open Science Framework (OSF).

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
