# Peer review of "Attitude and Performance for Online Learning during COVID-19 Pandemic: A Meta-Analytic Evidence"

_ijerph, 2022, doi:10.3390/ijerph191912967_

Round 1

Reviewer 1 Report

It is considered that the article starts from offering a poorly formulated title in two aspects: (1) the negative expression when it should be done in a neutral way about the purpose of the research, not about the result itself; and (2) avoid repetition of words in the title, in this case “learning”, it is recommended to eliminate the first time. The document offers some inaccuracies from the Abstract, for example, it is not understood when it indicates that the study includes the following countries: China, Indonesia, American (it is not understood what this is), Nepal, Sri Lanka and Canada, too many countries to reach to identify 40 papers of n = 98,558, being a sample too small for countless countries that are not analyzed in the data at any time, in addition to the fact that in current times there is an excess of published information, also in the Abstract it does not specify the methodology followed in the study, so it lacks scientific solidity in the rest of the document. The text lacks a theoretical framework between the introduction and materials and methods. It presents an unnecessary geospatial analysis, which focuses only on the representation of China, the sample being so small that it becomes insignificant information. Data analysis is not sufficient or consistent, describes the meta-analysis methodology, but data analysis is not sufficient to achieve an objective not stated in the methodology. Regarding the References section, many of them are from recent editions, it should be considered to include the DOI data, in addition to the fact that fundamental data is missing in several, for example: Adnan, Akuratiya, Alexander and many more, where it is impossible to determine if these are books or electronic documents.

Author Response

26-Sep-2022

RE: Attitude and performance for online learning during COVID-19 pandemic: A meta-analytic evidence (ijerph-1931045R1)

Reply to Reviewers

We deeply appreciate professional referees for reviewing our work and for all the new insightful and constructive comments you shared for us, especially in the methodological risks, data analysis and reference documents. We have read them carefully, and do thank you for these helpful and valuable suggestions. Following these advice, we have revised this manuscript thoroughly, one-by-one. All the revisions have been marked by blue fonts in both response letter in favor of re-reviews. We do believe that the revised manuscript has reaped huge fruits from these insightful and constructive comments that you shared. Please see specific response and modifications we have made underneath.

# Reviewer 1

- 1. It is considered that the article starts from offering a poorly formulated title in two aspects: (1) the negative expression when it should be done in a neutral way about the purpose of the research, not about the result itself; and (2) avoid repetition of words in the title, in this case “learning”, it is recommended to eliminate the first time.

Response: We sincerely appreciate you for this practical and helpful suggestion, and sorry to you for the inappropriate expression in title. Following your kind advice, as aforementioned, we have changed the title of the article to “Attitude and performance for online learning during COVID-19 pandemic: A meta-analytic evidence.”

- 2. The document offers some inaccuracies from the Abstract, for example, it is not understood when it indicates that the study includes the following countries: China, Indonesia, American (it is not understood what this is), Nepal, Sri Lanka and Canada.

Response: We truly apologize to confuse you as these inaccurate statements. Here, we wanna report what sample populations are for these included studies, and demonstrated that samples derived from these countries mentioned above. Following your kind advice, we have removed these incorrect contents.

- 3. Too many countries to reach to identify 40 papers of n = 98,558, being a sample too small for countless countries that are not analyzed in the data at any time.

Response: This is a fairly crucial and helpful comment worthy to be clarified with more details, and we do appreciate you for raising this insightful comment. We fully concur with you for this consideration, and are aware of wrongly claiming these countries in this manuscript. Following that, we no long reported sample population information to avoid potential misunderstanding for readers, such as “We performed a meta-analysis for literature concerning attitudes and performance towards online learning during COVID-19 pandemic. Forty published papers (n = 98,558) were screened eligible for formal analysis.” In addition, according to this crucial concern you mentioned, we have warrant some cautions in the limitation section in the revised manuscript to state as follow: “One shortcoming worthy to note is that these conclusions should be extended warily as the biased sample populations”. Again, we do appreciate you for this crucial and helpful comment, and have drawn conclusions in the current study with more prudent tones throughout whole manuscript.

- 4. In addition to the fact that in current times there is an excess of published information

Response: We earnestly thank you for this valuable comment. To obviate overreaching conclusions, we have rephrased all the conclusions in the revised manuscript by using more prudent claims and statements. Furthermore, we quite concur with you for this consideration with regards to more information on the online learning satisfaction. Indeed, articles describing the online learning attitude and performance are far more than the 40 papers we included. However, according to inclusion criteria we predefined, we rigorously screened studies that examined learning attitudes and performance with standardized and quantitative measures during COVID-19 pandemic. Despite a considerable number of relevant studies, they reported results by using unstructured investigation or relying on non-standardized self-reporting. To provide robust evidence, we have to removed a portion of studies investigating this topic with neither non-quantitative way nor non-standardized measures. Though, we do believe this point is highly valuable to be mentioned, and thus added relevant discussion in revised manuscript as follow: Lastly, to provide robustly statistical evidence, we have to remove a certain number of relevant studies because of lacking standardized or quantitative measures. Thus, the total sample size may be limited in the current study. 

- 5. Also in the abstract it does not specify the methodology followed in the study, so it lacks scientific solidity in the rest of the document.

Response: Many thanks to you for raising these helpful concerns. We apologized for providing incomplete methods in the abstract section due to word limit. Following your most promising proposal, we are added the methodological details in this section to strengthen scientific solidity. Please see underneath for more details:

“The COVID-19 pandemic prominently hit almost all the aspects of our life, especially in routine education. For public health security, online learning has to be enforced to replace classroom learning. Thus, it is a priority to clarify how these changes bring about for students. To do so, we build random-effect model of meta-analysis to pool individual effect size for these published articles concerning the attitudes and performance towards online learning. Databases included Google Scholar, PubMed and (Chinese) CNKI repository. Further, moderated analysis and meta-regression were further used to clarify potential heterogenous factors impacting this pooled effect. Forty published papers (n = 98,558) were screened eligible for formal analysis. Meta-analytic results demonstrated that 13.3% (95% CI: 10.0 - 17.5) students possessed negative attitudes for online learning during COVID-19 pandemic. 12.7% (95% CI: 9.6 - 16.8) students were found to report poorer performance in online learning compared classroom learning. Moderated analysis revealed that pandemic period significantly moderated performance for online learning, with poor performance in early pandemic (P = 0.006). Results for meta-regression analysis showed that the negative attitudes could predict poor learning performance significantly (p = 0.026). In conclusion, online learning that caused by COVID-19 pandemic may bring about negative learning attitudes and poorer learning performance compared to face-to-face classroom learning, especially in early pandemic.” Thanks to you again for this practical comment.

- 6. The text lacks a theoretical framework between the introduction and materials and methods.

Response: Many thanks to you for coming up with this crucial and helpful comment. We are sorry to provide unclear theoretical structures linking introduction and methods/materials section. Here, please allowed us to explain how we conducted these analyses to address our theoretical assumptions (see Figure R1). COVID-19 pandemic enforced almost all the schools to transfer classroom education into online learning. However, the impact of this educational policy to these students are under ongoing debates. Despite substantial studies, contradictory results were obtained for attitudes towards this change (i.e., positive v.s. negative). Furthermore, the impact of enforcing online learning on academic performance is still controversial (improved v.s. decreased). In this vein, to tackle them well, we conducted meta-analysis to pool effect size so as to reveal what influences of online learning on attitudes and performance are during COVID-19 pandemic. Furthermore, as we briefly described in Introduction section, the reasons to why enforcing online learning leads to different impacts on students’ attitudes and performance are in two-fold: (1) educational level (e.g., less impact on college students compared to primary-secondary schools students); (2) pandemic stage (less impacts on later pandemic period compared early one). Thus, to clarify whether the disparities are caused by these factors, we further carried on moderated analysis. Finally, we performed an exploratory meta-regression analysis to examine whether such attitudes for online learning may be associated with ensuing performance. We expected that this analysis may add evidence to reveal the impact of attitudes towards online learning on performance during COVID-19 pandemic. Following this crucial advice, we have added more descriptions to streamline the theoretical framework and to tie it well with Methods/Materials section. Again, many thanks to you for this crucial comment, and we do believe these revisions are highly imperative.

----- Please see the Figure R1 in the attachment -----

Figure R1 Diagram to streamline how to link theoretical basis to methodological design.

- 7. It presents an unnecessary geospatial analysis, which focuses only on the representation of China, the sample being so small that it becomes insignificant information.

Response: Many thanks to you for raising this point out. Following your comment, we have removed this analysis from revised manuscript.

- 8. Data analysis is not sufficient or consistent, describes the meta-analysis methodology, but data analysis is not sufficient to achieve an objective not stated in the methodology. 

Response: We sincerely appreciate you for coming up with this practical and helpful comment. We are sorry to confuse you as we provided incomplete reports for methodological framework. Please allow us to structurally detail what analyses we have done to achieve our research purposes. As we framed above, there are four research purposes to be probed in the current study. For public health security, online learning become mainstream to replace face-to-face classroom learning during COVID-19 pandemic. However, the impacts of this change on students are controversial, whilst reports for such impacts are still contradictory, especially in learning attitudes (positive v.s. negative) and performance (improved v.s. impeded).

Thus, the first and second research purpose is to solve this debate - that is - revealing what attitudes/performance are for online learning during COVID-19 pandemic. To do so, we utilized meta-analytic model to pool individual effect size for each article that aimed to measure students’ attitudes/performance on online learning during COVID-19 pandemic. Meta-analytic model could provide a robustly statistical evidence to probe what true effects are by controlling between-study random and sampling errors. Thus, this methodological design (i.e., meta-analytic model) is suitable to examine the first research purpose in the current study.

As briefly stated in introduction section, there are two confounding factors leading to contradictory results for the attitudes and performance on the online learning during COVID-19 pandemic, that are, educational level (primary-secondary school v.s. college) and pandemic period (early v.s. later). In this vein, to validate whether the different results that found in previous studies are confounded by these factors, we used moderated-effect analysis. The moderated-effect model is designed to examine whether the pool effect size that estimated from meta-analysis is moderated by potential heterogenous factors. Thus, this model could be eligible to fulfil the second research purpose.

As for the fourth research purpose, we gonna explore whether the the attitudes for online learning could predict ensuing performance. This analytic part is based on an exploratory assumption - that is - the performance in online learning may be impacted by negative attitudes. In previous meta-analytic models, we estimated individual effect size for each study on both attitudes and performance. Thus, we tried to build the meta-regression model fitting the individual effect size of attitudes for performance. Meta-regression model could statistically test the linear association between two variables that estimated from meta-analytic models, which has been validated broadly in previous studies (e.g., Baker et al., 2009, International journal of clinical practice; Tompson and Hioggins, 2002, Statistics in medicine). Thus, this is why we adopted meta-regression model to examine the third research purpose in the current study.

Following this crucial comment you raised, we recognized that the incomplete reports for methodology let readers hard to follow what issues are addressed in the current study. Thus, to clearly state justification of methodology, we have added more details into the Method and Materials section. Please see underneath:

“Meta-analysis were implemented by using Comprehensive Meta-Analysis (CMA) software (V3). To estimate study-within and study-between heterogeneity in meta-analytic models, the I2 test and Cochran’s Q test were used. As indicated in previous studies, meta-analysis with fixed-effect model was suitable for estimating meta-analytic effect with no heterogeneity (I2 < 50%, P value ≥ 0.1), while random-effect model should be used to adjust high data study-within and study-between heterogeneity (I2 > 50%, P value<0.1). In the current study, the random-effect meta-analytic models were built to examine the attitudes and performance for online learning during COVID-19 pandemic due to high heterogeneity (see Results section). In addition, to validate whether these meta-analytic effects are moderated by potential factors (i.e., educational level and pandemic period), the moderated-effect models were constructed by labeling individual effect size into subgroups (i.e., primary-secondary school group and college group; early pandemic group (Feb, 2020 - Dec, 2020) and late group(Jan, 2021 - June, 2022)). Furthermore, for testing potential role of attitudes on performance for online learning, we build meta-regression model by fitting individual effect size of attitudes to performance for all the included studies. Finally, as quality control, funnel charts were generated for visually inspecting publication bias in the current study, with asymmetric distribution for high publication bias. In addition, the Kendall’s test and Egger’s test were also used for quantitative validation for publication bias , with p < 0.05 indicating significant publication bias."

- 9. Regarding the References section, many of them are from recent editions, it should be considered to include the DOI data, in addition to the fact that fundamental data is missing in several, for example: Adnan, Akuratiya, Alexander and many more, where it is impossible to determine if these are books or electronic documents.

Response: Many thanks to you for raising this kind suggestion. We have embedded DOI into each reference in the reference list.

Reviewer 2 Report

In the era of covid-19, online learning has always been considered the most effective way of learning, and even in post-pandemic era, it is still considered a learning strategy that can assissted face-to-face learning. However, compared with previous studies, this study puts forward different insights. The analysis results show that during the COVID-19 pandemic, many students have a negative attitude towards online learning, and even compared to classroom learning, the results of online learning  are even lower, showing that there are still some problems with online learning. The following are suggested contents:

1. The feelings of online learning for primary and secondary school students and college students are different, and there may even be gaps between primary and secondary students. Therefore, it is recommended to subdivide the education level and you may find that the attitude towards online learning will be very different at different educational levels.

2. The discussion chapter mentioned that the level of national income will affect students' satisfaction with online learning, but lack of resources will lead to lack of learning. Is it fair to analyze objects with similar learning resources?

3. This study mentions that the vast majority of students still report no significant improvement or satisfaction with online learning. It is believed that this situation can be partially explained by two reasons: social isolation and gaps in learning resources. May I ask the above conclusion is the summed up according to opinions of students, or  this study? If so, how did the this study come to such an argument?

Author Response

26-Sep-2021

RE: Attitude and performance for online learning during COVID-19 pandemic: A meta-analytic evidence (ijerph-1931045R1)

Reply to Reviewer

We deeply appreciate professional referee for reviewing our work and for all the insightful and constructive comments you shared for us, especially in the results presentation and discussion. Following these advice, we have revised this manuscript thoroughly, one-by-one. All the revisions have been marked by blue fonts in both response letter in favor of re-reviews. We do believe that these revisions are imperative and helpful to strengthen our work. Please see specific response and modifications we have made underneath.

- 1. The feelings of online learning for primary and secondary school students and college students are different, and there may even be gaps between primary and secondary students. Therefore, it is recommended to subdivide the education level and you may find that the attitude towards online learning will be very different at different educational levels.

Response:  We sincerely appreciate you for this insightful and interesting suggestion. Following that, we have attempted to do so as you recommended. However, the small sample size (k = 1 for primary students; k = 6 for secondary studies; k = 2 for combining primary and secondary students) does not meet the statistical prerequisite for building meta-analytic model. In this vein, we regret that the more specific roles of educational levels on this meta-analytic effect cannot be revealed in the current study. Though, we still appreciate you for this insightful idea, and believe that it is worthy to be probed in the future.

- 2. The discussion chapter mentioned that the level of national income will affect students' satisfaction with online learning, but lack of resources will lead to lack of learning. Is it fair to analyze objects with similar learning resources?

Response: We are quite grateful to you for these valuable comments. We fully concur with you for this crucial concern. As you reckoned, we do not build a meta-analytic model to actually examine the impact of learning resources that gaped by economic level on attitudes and performance for online learning during COVID-19 pandemic. This statement is largely based on self-reported results for these students, with respect to what factors are for bringing negative attitudes and poor performance (see Table R1). Results demonstrated that lacking learning resource is one of the main reason to impede satisfaction and performance for online learning. Thus, we tried to infer the gaps of learning resource may be a major factor to incur negative learning attitudes and poor performance for online learning. As we discussed in the manuscript, the economic gap was highlighted oftentimes to sharpen educational inequality (i.e., online learning resources) in COVID-19 crisis (Blundell et al., 2022, Annual Review of Economics; Meier Jæge and Blaabæk, 2020, Research in Social Stratification and Mobility). This is why we inferred that the gap of learning resources that caused by economic levels may be one of the main reasons to result in negative attitudes and poor performance for online learning.

Table R1 Summary for self-reported reasons to negatively evaluate online learning from previous studies.

Reason

Frequency

Reference

learning resource

(lack of learning materials, technical support, learning platform, learning equipment. limited access to the internet through a mobile phone or other deice)

28

(Xiong and Wu, 2020), (Shi et al., 2022), (J. Chen and Cui, 2020), (Guo and Wen, 2021), (Y. Liu and Zhang, 2020), (H. Yang and Li, 2021), (Ministry Of Education and The Chinese Physical Society, 2020), (Qin et al., 2020), (Mi et al., 2021), (Juntang Li et al., 2022), (Xia, 2020), (X. Sun et al., 2021), (Zhou, 2021), (Y. Zhang et al., 2022), (B. Wang et al., 2021), (Y. Yang, 2022), (M. Yang and Qiu, 2021), (Y. Wang, 2021), (Jiang, 2021), (Zou, 2021), (Bao, 2021), (Sim et al., 2021), (Syam and Achmad, 2022), (Sharma et al., 2020), (Akuratiya and Meddage, 2020), (Hamid et al., 2020a), (Lemay et al., 2021), (Bestiantono et al., 2020)

Social isolation

21

(Xiong and Wu, 2020), (Shi et al., 2022), (J. Chen and Cui, 2020), (Y. Liu and Zhang, 2020), (H. Yang and Li, 2021), (Ministry Of Education and The Chinese Physical Society, 2020), (Qin et al., 2020), (Mi et al., 2021), (Xia, 2020), (Zhou, 2021), (He et al., 2022), (Y. Zhang et al., 2022), (Y. Yang, 2022), (M. Yang and Qiu, 2021), (Jiang, 2021), (Sim et al., 2021),  (Syam and Achmad, 2022), (Akuratiya and Meddage, 2020), (Alawamleh et al., 2020), (Lemay et al., 2021), (Bestiantono et al., 2020)

Teacher behavior

(teaching ability, preparation, supervision and teaching guide)

11

(Xiong and Wu, 2020), (Shi et al., 2022), (Y. Liu and Zhang, 2020), (H. Yang and Li, 2021), (B. Wang et al., 2021), (Y. Yang, 2022), (M. Yang and Qiu, 2021), (Y. Wang, 2021), (Jiang, 2021), (Zou, 2021), (Hamid et al., 2020a)

Learning autonomy

9

(Xiong and Wu, 2020), (Xia, 2020), (S. Sun and Wu, 2021), (Zhou, 2021), (He et al., 2022), (Y. Yang, 2022), (M. Yang and Qiu, 2021), (Syam and Achmad, 2022), (Lemay et al., 2021)

Teaching contents

(course arrangement, course design, course time and classroom activities)

8

(Y. Liu and Zhang, 2020), (H. Yang and Li, 2021), (Zhou, 2021), (Y. Zhang et al., 2022), (Y. Yang, 2022), (M. Yang and Qiu, 2021), (Y. Wang, 2021), (Jiang, 2021)

Following this crucial comment you shared for us, we are aware of overreaching conclusions from these findings in the current manuscript - that is - there is no direct evidence to substantiate the moderated roles of economic gap on the association between learning resource skewness and negative attitudes/performance for online learning. To tackle it well, we have rewritten this section in “more softer” tones for obviating overreaching conclusions. Please see specific details for what we have modified underneath: ... On the other hand, existing evidence was provided to indicate the potential impact of learning resource gap on learning satisfaction and fairness during COVID-19 pandemic (Abou Khalil et al., 2021; Pather et al., 2020). Specifically, compared to middle- and low-income countries (areas), students in high- or upper-middle-income countries (areas) rated online learning for more positive ranks due to superior learning resources (Faize and Nawaz, 2020; Qazi et al., 2020). Supporting that, further studies revealed that students with disadvantaged positions in economic would receive less high-quality learning resource so as to incure poor academic performance (Alexander et al., 2020; Areba and Ngwacho, 2020; Blackman, 2021). Thus, these indirect evidence may provide another insight to warrant stakeholders for the potential risk of this change (i.e., enforcing online learning) in exacerbating educational inequality. In addition, it is remarkable that this conjecture should be further validated elsewhere as lacking direct evidence in the current study.

- 3. This study mentions that the vast majority of students still report no significant improvement or satisfaction with online learning. It is believed that this situation can be partially explained by two reasons: social isolation and gaps in learning resources. May I ask the above conclusion is the summed up according to opinions of students, or  this study? If so, how did the this study come to such an argument?

Response: We are quite grateful to you for raising this crucial question. These conclusions mentioned above are grounded on actual self-reports from students in summing up these literature. To probe into why students seem to dissatisfy online learning (compared to face-to-face classroom learning), the systematic literature summary for self-reported reasons on dissatisfaction towards online learning in these students has been conducted. Results demonstrated that the learning resource gap and social isolation predominated reasons to dissatisfy online learning during COVID-19 pandemic (please see Table R1 for details). This is why we raised such points to explain why this change (i.e., online learning) that caused by COVID-19 pandemic may bring about negative impacts for students.

As this section is go outside of scope in the current quantitative meta-analytic evidence, we did not report these results in main texts previously. According to this crucial point you commented, we have provided these details in the revised manuscript to justify how we come up with these opinions. Again, many thanks to you for these crucial and helpful comments. We do believe that these advice have reaped huge fruits for the current study.

Round 2

Reviewer 1 Report

The authors correctly addressed the observations previously made